# The Diagnostic Performance of 2-[^18^F]FDG PET/CT in Identifying Richter Transformation in Chronic Lymphocytic Leukemia: An Updated Systematic Review and Bivariate Meta-Analysis

**DOI:** 10.3390/cancers16091778

**Published:** 2024-05-05

**Authors:** Domenico Albano, Alessio Rizzo, Manuela Racca, Barbara Muoio, Francesco Bertagna, Giorgio Treglia

**Affiliations:** 1Nuclear Medicine, University of Brescia and ASST Spedali Civili Brescia, 25123 Brescia, Italy; francesco.bertagna@unibs.it; 2Department of Nuclear Medicine, Candiolo Cancer Institute, FPO-IRCCS, 10060 Turin, Italy; alessio.rizzo@ircc.it (A.R.); manuela.racca@ircc.it (M.R.); 3Division of Medical Oncology, Oncology Institute of Southern Switzerland, Ente Ospedaliero Cantonale, 6501 Bellinzona, Switzerland; barbara.muoio@eoc.ch; 4Clinic of Nuclear Medicine, Imaging Institute of Southern Switzerland, Ente Ospedaliero Cantonale, 6500 Bellinzona, Switzerland; giorgio.treglia@eoc.ch; 5Faculty of Biomedical Sciences, Università della Svizzera Italiana (USI), 6900 Lugano, Switzerland; 6Department of Nuclear Medicine and Molecular Imaging, Lausanne University Hospital, University of Lausanne, 1011 Lausanne, Switzerland

**Keywords:** FDG, PET/CT, systematic review, meta-analysis, SUV, CLL, Richter transformation

## Abstract

**Simple Summary:**

Richter transformation (RT) is characterized by the transformation of CCL in the aggressive lymphoma variant with a significant worsening in prognosis. Initial reports about a substantial impact of 2-deoxy-2-[^18^F]-fluoro-D-glucose positron emission tomography/computed tomography (2-[^18^F]FDG PET/CT) in predicting RT are available in the literature. Using data from 15 published studies, including 1593 CLL patients, we demonstrated that 2-[^18^F]FDG uptake expressed as the maximum standardized uptake value (SUV_max_) has a high negative predictive value.

**Abstract:**

Richter transformation is a rare phenomenon characterized by the transformation of cell chronic lymphocytic leukemia (CLL) into a more aggressive lymphoma variant. The early identification of CLLs with a high risk of RT is fundamental. In this field, 2-deoxy-2-[^18^F]-fluoro-D-glucose positron emission tomography/computed tomography (2-[^18^F]FDG PET/CT) has been shown to be a non-invasive and promising tool, but apparently, unclear data seem to be present in the literature. This systematic review and bivariate meta-analysis aimed to investigate the diagnostic performance of 2-[^18^F]FDG PET/CT and its parameters in predicting RT. Between 2006 and 2024, 15 studies were published on this topic, including 1593 CLL patients. Among semiquantitative variables, SUV_max_ was the most investigated, and the best threshold derived for detecting RT was five. With this cut-off value, a pooled sensitivity of 86.8% (95% CI: 78.5–93.3), a pooled specificity of 48.1% (95% CI: 27–69.9), a pooled negative predictive value of 90.5% (95% CI: 88.4–92.4), a pooled negative likelihood ratio of 0.35 (95% CI: 0.17–0.70), a pooled positive likelihood ratio of 1.8 (95% CI: 1.3–2.4), and a pooled diagnostic odds ratio of 6.7 (3.5–12.5) were obtained. With a higher cut-off (SUV_max_ = 10), the specificity increased while the sensitivity reduced. The other metabolic features, like metabolic tumor volume, total lesion glycolysis, and radiomic features, were only marginally investigated with controversial evidence.

## 1. Introduction

Chronic lymphocytic leukemia (CLL) is a form of leukemia typical of elderly people and with a variable clinical course [1]. The pathogenesis of CLL is characterized by the clonal expansion of CD5+CD23+ B cells in blood, marrow, and second lymphoid tissues described as lymph nodes and spleen [2]. Normally, CLL is a less aggressive lymphoproliferative disease with an optimal prognosis, but in some instances (less than 10%), it transforms into a more aggressive condition associated with poor outcomes. This condition is called Richter transformation (RT) [3] and was described for the first time by Maurice Richter, a pathologist, in 1928. Usually, CLL evolves into diffuse large B-cell lymphoma (DLBCL) and, less frequently, into Hodgkin lymphoma (HL) [4,5].

The prognosis of CLL with RT is poor due to the absence of efficient treatments despite recent improvements [6,7,8]. It is fundamental to diagnose this transformation as soon as possible. The best way to detect RT is histological confirmation performed through an excisional node biopsy or a core needle biopsy. Of course, it is crucial to detect the correct lymph node for biopsy [9]. It was demonstrated that the tumor size is not the best criterion for the choice of which lymph node to investigate because morphological information lacks functional data. In this context, emerging data about the potential role of 2-deoxy-2-[^18^F]-fluoro-D-glucose positron emission tomography/computed tomography (2-[^18^F]FDG PET/CT) in the choice of the lymph nodes for histological examination are present. 2-[^18^F]FDG PET/CT is a non-invasive imaging technique with high accuracy in recognizing the aggressiveness and risk of evolution of lymph nodes in CLL [10,11]. The standardized uptake value (SUV) is a semiquantitative PET parameter that indirectly expresses the metabolic activity in a specific area reflecting glucose metabolism. Usually, CLL is a disease with low/moderate 2-[^18^F]FDG uptake; thus, an increased uptake expressed as high SUV could be an indirect expression of aggressive transformation, characteristic of RT. The current body of literature suggests different SUV thresholds to discriminate patients with RT.

This updated systematic review and bivariate meta-analysis aims to analyze the published findings about the diagnostic performance of 2-[^18^F]FDG PET/CT and its parameters in recognizing RT.

## 2. Materials and Methods

### 2.1. Protocol

The current systematic review was carried out following a preset protocol, and the “Preferred Reporting Items for a Systematic Review and Meta-Analysis” (PRISMA 2020 statement) served as a guideline for its development and reporting. The protocol has not been registered. The complete PRISMA checklist can be found in Appendix A. As a first step, a direct review query was expressed: “What is the diagnostic performance of 2-[^18^F]FDG PET or PET/CT in detecting RT of CCL?” Following the Population, Intervention, Comparator, and Outcomes (PICO) framework, we defined the criteria for study inclusion: studies performed on patients affected by CLL and with suspected RT (Population), undergoing PET or PET/CT with 2-[^18^F]FDG (Intervention) compared or not with standard-of-care imaging (Comparator). The primary outcome was the diagnostic performance of 2-[^18^F]FDG PET or PET/CT in recognizing RT. Two investigators (A.R. and D.A.) independently performed the literature search, study selection, data extraction, and quality evaluation.

### 2.2. Search Strategy

A comprehensive literature search of the PubMed/MEDLINE, Embase, and Cochrane library databases was conducted to extract relevant published articles about the diagnostic performance of 2-[^18^F]FDG PET or PET/CT in detecting RT. The ClinicalTrials.gov database was additionally searched for ongoing investigations (access date: 1 March 2024). We created and used a search algorithm based on a combination of the terms (a) “chronic lymphocytic leukemia” OR “CLL” OR “Richter transformation” OR “Richter syndrome” AND (b) “positron emission tomography” OR “PET” AND (c) “FDG” OR “fluorodeoxyglucose”. No start date limit was used; the search was updated until 31 March 2024. In order to enlarge our analysis, the retrieved articles’ references were also screened for searching additional articles related to the topic of interest. 

### 2.3. Study Selection 

Original articles or subsets in studies focused on the role of 2-[^18^F]FDG PET or PET/CT in recognizing RT in patients with CLL were eligible for inclusion in this systematic review. The exclusion criteria of our research were as follows: (a) studies not in the field of interest (including preclinical studies); (b) case reports or small case series (less than five patients with RT events); and (c) non-original studies like review articles, meta-analyses, editorials, letters, and conference proceedings. No language restriction was used. Two researchers (D.A. and A.R.) independently reviewed the titles and abstracts and read the full manuscripts of the retrieved articles, applying the criteria described above. In case of disagreements, a third researcher (GT) was consulted.

### 2.4. Data Extraction and Collection

To avoid potential biases, the researchers separately gathered each of the studies and extracted data from the information in the entire manuscript, figures, and tables. For each included study, we collected data concerning overall study information (authors, year of publication, country, study design, founding sources, number of included subjects, gender, age, number of RTs) and technical variables (PET device used, administered radiopharmaceutical, kind of hybrid imaging procedure, administered activity, uptake time, image analysis features).

Diagnostic performances expressed as sensitivity, specificity, positive predictive value (PPV), and negative predictive value (NPV) of 2-[^18^F]FDG PET or PET/CT in detecting RT in patients with CLL were derived. The principal data of the articles included in this review are described in Tables and in the Section 3.

### 2.5. Quality Assessment (Risk of Bias Assessment) 

A quality assessment of the included articles was performed to analyze the risk of bias in individual studies and their relevance to the review query. Four areas (patient selection, index test, reference standard, and flow and timing) were assessed for risk of bias. Moreover, three aspects were assessed for applicability concerns (patient selection, index test, and reference standard) by using the QUADAS-2 tool [12]. Two researchers (A.R. and D.A.) independently evaluated the quality of the included studies in the systematic review.

### 2.6. Statistical Analysis 

For the calculation of diagnostic accuracy, we performed a per-patient-based analysis in each study, extracting, if possible, true positive, true negative, false positive, and false negative reports. In the quantitative analysis, the pooled sensitivity and specificity were used as main outcome values, and a bivariate random-effects model was applied to calculate these metrics. The bivariate random-effects model takes into account the possible correlation between sensitivity and specificity [13]. Then, we calculated pooled negative and positive likelihood ratios (LR− and LR+) and diagnostic odds ratios (DORs). Pooled outcome measures were provided with 95% confidence interval values (95%CIs). We used a summary receiver operating characteristic (SROC) curve to recap the diagnostic performance of the index test [13]. In case of significant statistical heterogeneity, subgroup analyses were planned, considering patient characteristics, index test features, clinical scenarios, and technical aspects. The inconsistency index (I^2^ index or I-square) was used to assess the presence of statistical heterogeneity (with significant heterogeneity present for I^2^ values > 50%) [13]. Egger’s test was tested for the evaluation of publication bias. We used OpenMeta Analyst® (Brown University, Providence, RI, USA, version 10.12) for all statistical analyses.

## 3. Results

### 3.1. Literature Search

After the literature search from the selected databases, we derived 135 records. After reading the titles and abstracts, 120 articles were excluded due to them being not in the field of interest (*n* = 60), reviews or editorials (*n* = 20), case reports or small case series (*n* = 36), and preclinical studies (*n* = 4). Lastly, 15 studies were screened in the full-text version, and all were included in this systematic review [14,15,16,17,18,19,20,21,22,23,24,25,26,27,28] (Figure 1). No additional manuscripts were added after the revision of the references of the selected records. The Appendix A) list all the excluded studies.

### 3.2. Study Characteristics

The fifteen studies that met the criteria for inclusion in the systematic review (qualitative analysis) are carefully analyzed in Table 1, Table 2, and Table 3. A total of 1593 CLL patients were included. Most studies were retrospective [14,15,16,19,20,21,23,24,25,26,27,28] except for three with a prospective design [17,18,22]. The selected articles were published between 2006 and 20024 in the USA (6/15), Italy (5/15), France (2/15), Israel (1/15), and the Czech Republic (1/15). Four studies disclosed financing resources in the text [18,22,24,25]. The median/mean age ranged from 57.7 to 71 years, and males were more prevalent than females in all studies except for one [21]. RT was registered in 320 patients with an average prevalence of 20% (range: 7–100%) (Table 1).

Thirteen articles employed PET/CT as a hybrid imaging device [14,16,17,18,20,21,22,23,25,26,27,28], two studies used PET scanners only [15,24], and one used both techniques (PET and PET/CT) [19]. The mean injected radiotracer activity was heterogeneous. When evaluated as relative values, the administered activity ranged from 3.5 to 4.5 MBq/Kg, while when calculated using absolute values, it ranged from 166 to 700 MBq. The elapsed time from injection to acquisition was about 60 minutes in all investigations except for one study, which reported 45 minutes [15]. PET or PET/CT images were analyzed visually and semiquantitatively in all the included studies. Regarding semiquantitative parameters, the maximum standardized uptake value (SUV_max_) was the most common PET feature measured, followed by metabolic tumor volume (MTV) and total lesion glycolysis (TLG) [23,26,28]. Radiomic features were investigated only in one included study [28].

### 3.3. Risk of Bias and Applicability 

The overall estimation of the risk of bias and concerns regarding the applicability of articles included in the systematic review according to QUADAS-2 are summarized in Figure 2.

### 3.4. Primary Results of the Included Studies (Qualitative Synthesis)

In all studies, the site of biopsy was described as the site of highest uptake (with the highest SUV_max_). For the detection of RT, both qualitative and semiquantitative PET image analyses were performed. In five investigations [15,16,17,23,27], only visual analysis of PET images was applied, and SUV_max_ was not measured (Table 3). In the other studies [14,18,19,20,21,22,24,25,26,28], semiquantitative parameters were extracted and tested to predict RT. SUV_max_ was the variable most frequently applied, and the cut-off value of five was the most commonly investigated alone [14,18,19,20,25,28] or alongside other values [22,24]. In one study [26], the threshold of SUV_max_ tested was nine; in another [21] it was ten. Among papers testing five as the SUV_max_ cut-off, a mean overall sensitivity of 87% (range: 71–96%), a mean overall specificity of 49% (range: 4–80%), a mean overall PPV of 41% (range: 16–53%), and a mean overall NPV of 84% (range: 33–97%) were reported. Instead, sensitivity, specificity, PPV, and NPV in the study with an SUV_max_ cut-off of nine were 67%, 90%, 67%, and 90%, and 91%, 95%, 29%, and 99% in publication with an SUV_max_ cut-off of ten. In two studies [22,24], different SUV_max_ cut-off values were compared, and as the SUV_max_ value thresholds increased, the sensitivity decreased while specificity was augmented. In two studies [26,28], metabolic features different from SUV_max_ were investigated (such as SUV corrected for lean body mass (SUVlbm) and for body surface area (SUVbsa), the lesion-to-blood-pool SUV ratio (L-BP SUV R), the lesion-to-liver SUV ratio (L-L SUV R), total lesion glycolysis (TLG), and metabolic tumor volume (MTV)), demonstrating an ability to predict RT significantly with diagnostic performance similar to SUV_max_, except for MTV and TLG, which showed no significant role. Only one article [28] tested radiomic features, but no significant association with the ability to discriminate RT was demonstrated.

**Table 3 cancers-16-01778-t003:** Diagnostic outcome of 2-[^18^F]FDG PET with different SUV_max_ cut-offs.

Author	SUV_max_ Cut-Off	TP	FN	FP	TN	Sen	Spe	PPV	NPV
Bruzzi JF [14]	5	10	1	8	18	91%	69%	56%	95%
Karam M [15]	Na	na	na	na	Na	Na	na	Na	na
Taralli S [16]	Na	na	na	na	Na	na	na	Na	na
Papajik T [17]	Na	na	na	na	Na	na	na	Na	na
Conte MJ [18]	5	22	na	61	Na	na	na	Na	na
Falchi L [19]	5	84	11	125	112	88%	47%	38%	92%
Mauro FR [20]	5	15	2	24	49	88%	67%	38.5%	96%
Michallet AS [21]	10	22	11	2	205	92%	95%	66%	99%
Mato AR [22]	5 10 11 12	5 5	1 14	27 2	1 14	71% 71% 71% 57%	4% 50% 61% 68%	16% 26% 31% 31%	33% 88% 89% 86%
Pontoizeau C [23]	Na	na	na	na	Na	na	na	Na	na
Wang Y [24]	5 6 7 8 9 10 11 12 13 14 15	24 14	1 7	53 11	14 22	96% 92% 84% 76% 72% 56% 52% 44% 40% 28% 28%	21% 28% 45% 62% 72% 76% 83% 86% 93% 93% 93%	51% 52% 57% 63% 69% 67% 72% 73% 83% 78% 78%	86% 80% 76% 75% 75% 67% 67% 64% 64% 63% 60%
Porrazzo M [25]	5	4	1	9	26	80%	74%	31%	96%
Albano D [26]	9	12	6	6	61	67%	91%	67%	90%
Hod K [27]	Na	na	na	na	Na	na	Na	Na	na
Albano D [28]	5	32	14	32	73	70%	70%	50%	80%

TP: true positive; FN: false negative; FP: false positive; TN: true negative; Sen: sensitivity; Spe: specificity; PPV: positive predictive value; NPV: negative predictive value.

### 3.5. Quantitative Analysis (Meta-Analysis)

For the quantitative analysis, we investigated the role of the two most commonly employed SUV_max_ thresholds in predicting RT. Seven studies, including 773 patients with CCL, reported the performance of an SUV_max_ threshold of five to predict RT and were included in the bivariate meta-analysis [14,19,20,22,24,25,28]. Based on a per-patient analysis, the pooled sensitivity and specificity of 2-[^18^F]FDG PET for the assessment of RT were 0.868 (95% CI: 0.785–0.933) and 0.481 (95% CI: 0.270–0.699). The related SROC curve is shown in Figure 3. The pooled LR−, LR+, and DOR were 0.349 (95% CI: 0.175–0.696), 1.801 (95% CI: 1.351–2.402), and 6.689 (3.573–12.525), respectively (Figure 4 and Figure 5). There was significant statistical heterogeneity among the studies included in this analysis, as the inconsistency index was always higher than 50% in all tests except for the pooled DOR (less than 50%). Based on the statistical heterogeneity observed and as stated in the Section 2, the authors performed two subgroup analyses: the first omitting studies reporting fewer events [22,25] and the second excluding studies not using hybrid PET/CT in all or in a subset of patients [19,24], with similar results compared to the main analysis.

Four articles including 526 patients diagnosed with CCL reported the diagnostic performance of an SUV_max_ cut-off of 10 to predict RT and were included in the random-effects patient-based meta-analysis [21,22,24,28]. Based on a per-patient analysis, the pooled sensitivity and specificity of 2-[^18^F]FDG PET for the assessment of RT were 0.574 (95% CI: 0.383–0.745) and 0.912 (95% CI: 0.690–0.6980). The related SROC curve and forest plots are shown in Figure 6 and Figure 7, respectively. There was significant statistical heterogeneity among the studies included in this analysis, as the inconsistency index was always higher than 50%. Due to the constrained number of included studies in this analysis, a subgroup test was not feasible.

## 4. Discussion

The usefulness of 2-[^18^F]FDG PET/CT in studying lymphoproliferative diseases is well demonstrated, especially in the staging and treatment response evaluation of some lymphoma histotypes, such as Hodgkin lymphoma (HL), DLBCL, and FL [29]. One of the emerging indications of 2-[^18^F]FDG PET/CT is detecting the best site for biopsy to recognize a potential transformation of indolent lymphomas in more aggressive variants [30,31]. Usually, the site with higher metabolic tracer uptake is associated with a higher risk of aggressive evolution. For these reasons, in all studies, the biopsy was performed at the site of highest uptake (with highest SUV_max_). Of course, this confirmation is derived by histological analysis. Still, finding the best site to perform a biopsy is crucial, even more so if patients have plural nodal localizations, which are common in CCL patients. The morphological findings are not sufficiently accurate to detect the correct site for biopsy with the highest transformation rate. Since CLL is a disease usually characterized by mild 2-[^18^F]FDG uptake, this tracer seems to be an optimal probe to diagnose RT due to its ability to make lesions with increased proliferative activity evident, expressed as high glucose metabolism. Data in the literature clearly show that 2-[^18^F]FDG uptake is directly associated with disease aggressiveness and able to detect the transformation of indolent lymphomas (such as CLL) into aggressive lymphoma variants (RT) [14,15,16,17,18,19,20,21,22,23,24,25,26,27,28].

The most frequent PET parameter investigated to predict RT was SUV_max_. This feature is effortless to measure, fast, and reproducible in the same patient; however, it presents several limitations that reduce the reliability and reproducibility between different facilities and patients, such as the uptake time (time between injection and imaging scan), the size of the measured lesions (partial volume effect for small lesions), the risk of extravasation of the administered radiotracer, any residual activity of the radiotracer in the syringe, the decay of the injected activity, and some acquisition or reconstruction characteristics (type of tomograph, type of algorithm applied, artifacts) [32]. In other lymphomas, metabolic tumor burden and radiomic features were demonstrated to be superior to SUV_max_ [33,34,35,36], but this evidence was not confirmed in CCL patients [28].

The most frequent SUV_max_ cut-off investigated to predict RT was five. Among the included studies, this threshold was usually derived by empirical observation or by applying an ROC curve analysis. With this threshold, 2-[^18^F]FDG PET/CT showed very high sensitivity but low specificity, with a high risk of false positive findings but, interestingly, almost no false negative reports [14,15,16,17,18,19,20,21,23,24,25,26,28]. Thus, we could suggest using 2-[^18^F]FDG PET/CT to select patients who will benefit from not using an invasive approach, such as a biopsy. Instead, with the increase in the SUV_max_ threshold (i.e., 9 or 10), the sensitivity will decrease in favor of specificity due to a higher risk of false negative findings for RT detection [21,22,24,26].

In two studies [22,24], the authors compared different values of SUV_max_, confirming this “mirror” effect between SUV_max_ and diagnostic performances. In the first case [22], the false positive findings had shrunk from 26 to 2, with a consequent gain in the number of true negative cases. Similar results were obtained from the other study by Wang et al. [24].

The technological drift towards digital PET/CT scanners and novel radiopharmaceuticals might affect the management of these patients, particularly the role of SUV_max_. It is well demonstrated that digital PET imaging [37,38] and even long axial field of view (LAFOV) PET [39] have increased image quality and lesion detection rates compared to conventional scanners. Also, the semiquantitative analysis suffered substantial changes with an increase in coincidences, which might lead to higher values for SUV-based metrics. Among the included studies, none of the scholars reported the use of digital or LAFOV PET scanners. Still, it is presumable that the new SUV thresholds will be calculated with an even further increased diffusion of this technology.

Another development field is the investigation of alternative radiopharmaceuticals, like [^68^Ga]Ga-Pentixafor. Preliminary evidence underlines the potential superiority of this novel radiopharmaceutical over 2-[^18^F]FDG in CCL patients [40,41]. However, more solid evidence is needed to confirm or challenge this point of view, especially considering the potential employment of this compound as a theragnostic agent through its labeling with [^177^Lu]Lu.

Lastly, several limitations of this systematic review and meta-analysis should be underlined: first, the design of the studies, often retrospective and monocentric; second, the relatively low number of patients included in each study (directly related to the rare prevalence of RT); and third, the wide heterogeneity related to several factors including characteristics of the studies and patients included and aspects related to the index test or the target condition.

## 5. Conclusions

With limitations of the heterogeneity of the studies included, with this systematic review and meta-analysis, we can reason that 2-[^18^F]FDG PET/CT has a significant role in the detection of RT in CLL patients, especially showing high sensitivity and negative predictive values when applying an SUV_max_ threshold of five. Increasing the SUV_max_ value threshold resulted in a gain in specificity but at the expense of sensitivity. Due to the still-limited data, prospective and multicentric studies are warranted to validate these preliminary findings.

## Figures and Tables

**Figure 1 cancers-16-01778-f001:**
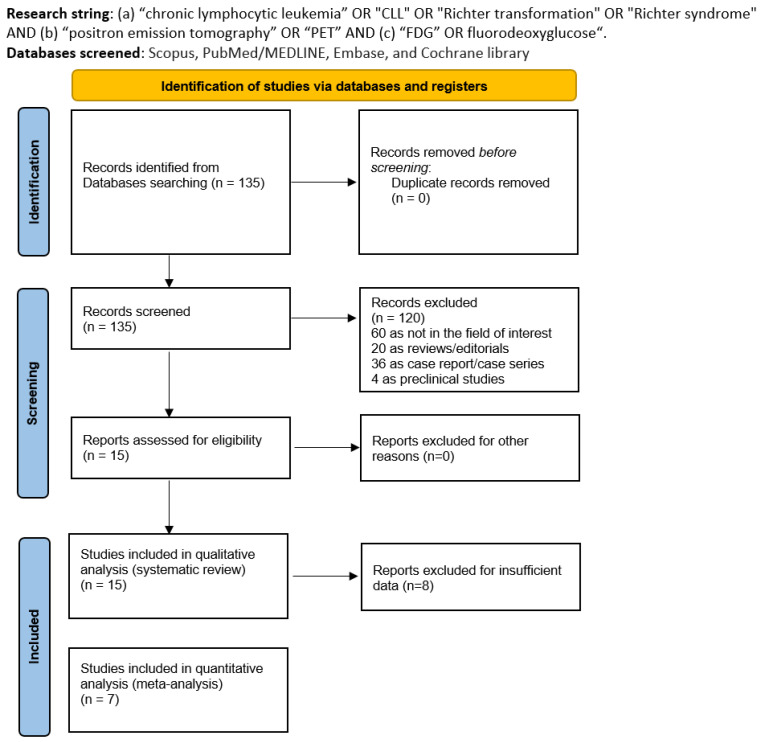
Literature search flowchart.

**Figure 2 cancers-16-01778-f002:**
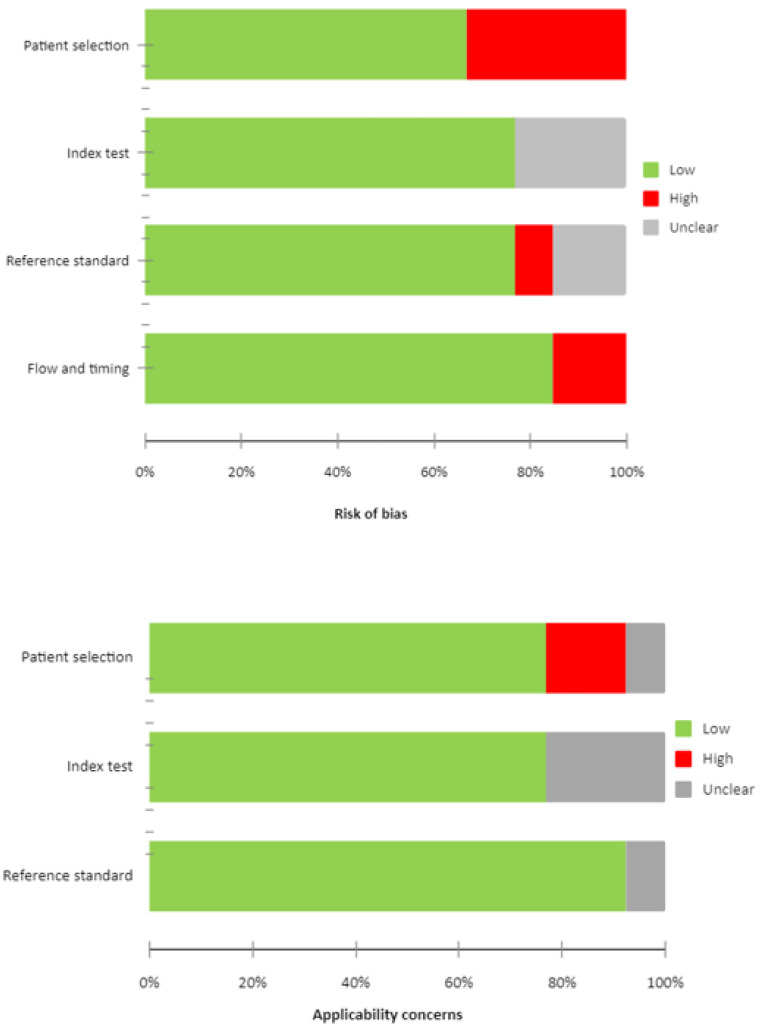
QUADAS 2 score of the articles.

**Figure 3 cancers-16-01778-f003:**
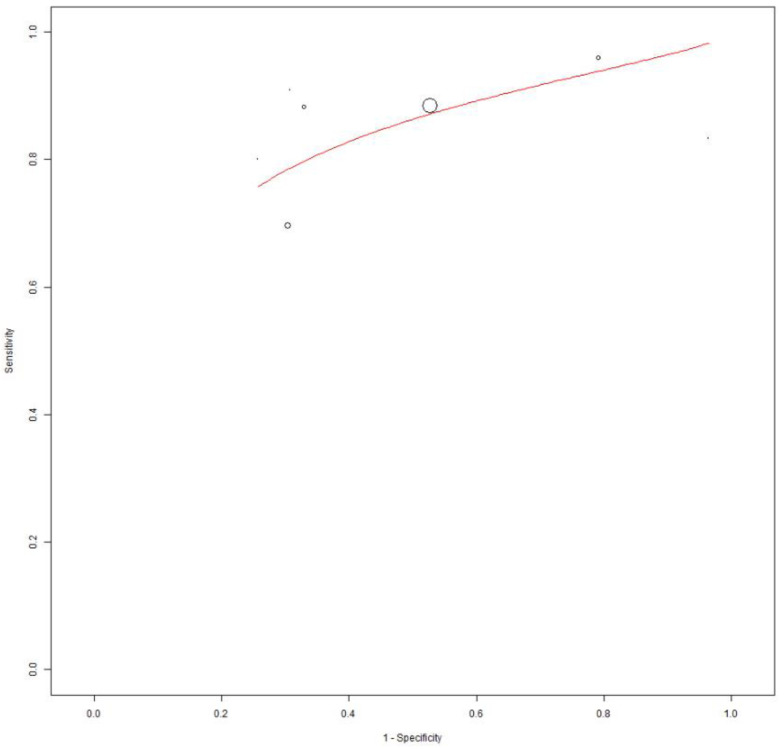
SROC curve of index test’s diagnostic accuracy using an SUV_max_ cut-off of 5 to predict RT.

**Figure 4 cancers-16-01778-f004:**
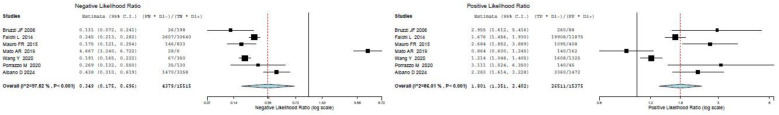
Pooled negative and positive likelihood ratios of the index test in the assessment of RT applying a SUV_max_ cut-off of 5 as the threshold [14,19,20,22,24,25,28]. Legend: 95% C.I.: 95% confidence interval; TP: true positive; FN: false negative; FP: false positive; TN: true negative.

**Figure 5 cancers-16-01778-f005:**
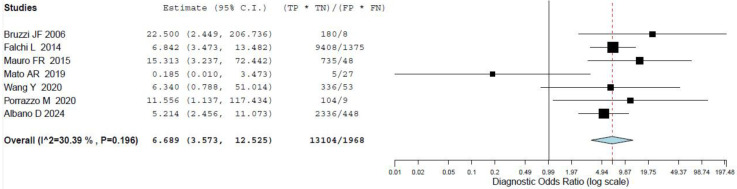
Pooled diagnostic odds ratio of the index test in the assessment of RT applying an SUV_max_ of 5 as threshold [14,19,20,22,24,25,28]. Legend: 95% C.I.: 95% confidence interval; TP: true positive; FN: false negative; FP: false positive; TN: true negative.

**Figure 6 cancers-16-01778-f006:**
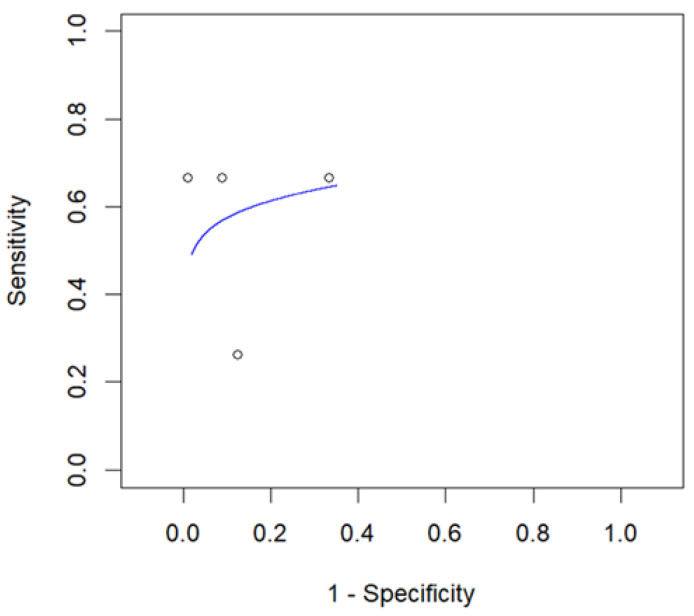
SROC curve of index test’s diagnostic accuracy using an SUV_max_ cut-off of 10 to predict RT.

**Figure 7 cancers-16-01778-f007:**
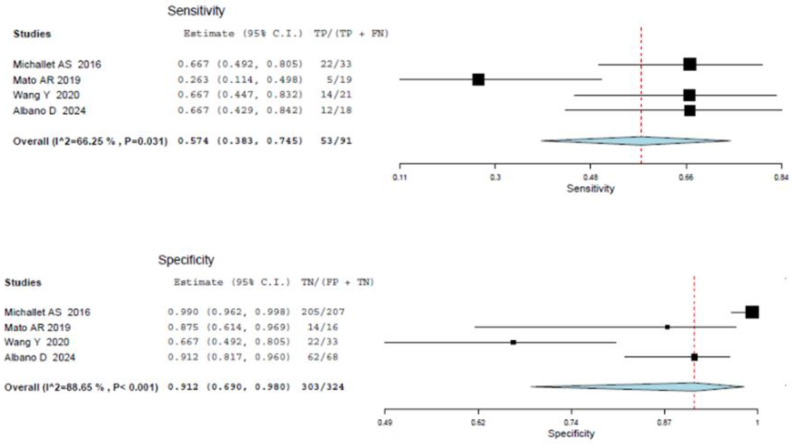
Sensitivity and specificity of the index test in the assessment of RT applying an SUV_max_ of 10 as cut-off [21,22,24,28]. Legend: 95% C.I.: 95% confidence interval; TP: true positive; FN: false negative; FP: false positive; TN: true negative.

**Table 1 cancers-16-01778-t001:** Basic study and patient characteristics.

First Author	Year	Country	Study Design	Funding Sources	No. of CLL Patients	M/F	Mean Age (Range)	No. of RT Patients
Bruzzi JF [14]	2006	USA	Retrospective	None declared	37	26:11	61 (40–82)	11 (30%)
Karam M [15]	2006	USA	Retrospective	None declared	15	NR	NR	1 (7%)
Taralli S [16]	2012	Italy	Retrospective	None declared	9	8:1	57.7 (49–70)	1 (11%)
Papajik T [17]	2014	Czech Republic	Prospective	None declared	44	NR	NR	8 (18%)
Conte MJ [18]	2014	USA	Prospective	Jackie S. Taylor Memorial Fund and the University of Iowa/Mayo Clinic Lymphoma SPORE (CA097274)	272	197:75	61.5 * (21–91)	25 (9%)
Falchi L [19]	2014	USA	Retrospective	None declared	332	218:114	68 * (31–85)	95 (29%)
Mauro FR [20]	2015	Italy	Retrospective	None declared	90	65:25	61.2 * (31–81)	17 (19%)
Michallet AS [21]	2016	France	Retrospective	None declared	240	94:146	62 (21–91)	24 (10%)
Mato AR [22]	2019	USA	Prospective	AbbVie, Inc. and Genentech-Roche, Inc.	57	Nr	67 * (28–85)	8 (14%)
Pontoizeau C [23]	2020	France	Retrospective	None declared	28	22:6	71 * (36–89)	28 (100%)
Wang Y [24]	2020	USA	Retrospective	K12 CA090628 grant from the National Cancer Institute (Paul Calabresi Career Development Award for Clinical Oncology)	92	69:23	68 * (43–89)	25 (27%)
Porrazzo M [25]	2020	Italy	Retrospective	Associazione Italiana per la Ricerca sul Cancro (AIRC) Foundation Milan, Italy, grant number (AIRC 5 _ 1000 No. 21198)	40	31:9	62 * (35–92)	5 (13%)
Albano D [26]	2021	Italy	Retrospective	None declared	80	58:22	61 (27–83)	18 (22.5%)
Hod K [27]	2021	Israel	Retrospective	None declared	120 °	72:48	64	8 (6.7%)
Albano D [28]	2024	Italy	Retrospective	None declared	137	103:34	62 *	46 (34%)

M: male; F: female; CLL: chronic lymphocytic leukemia; RT: Richter transformation; NR: not reported; *: median; °: among them, 17 small lymphocytic lymphoma.

**Table 2 cancers-16-01778-t002:** Index test key characteristics.

First Author	Hybrid Imaging	Tomograph	2-[^18^F]FDG Mean Injected Activity (MBq)	Uptake Time (min)	Image Analysis	Semiquantitative Parameters
Bruzzi JF [14]	PET/CT	Discovery ST-8, GE Healthcare	555	60	Visual and semiquantitative	SUV_max_
Karam M [15]	PET	Advance NXI, GE Healthcare	592–700	45	Visual and semiquantitative	SUV_max_
Taralli S [16]	PET/CT	GEMINI DUAL and GEMINI GXL, Philips Medical System	166–318	60	Visual and semiquantitative	SUV_max_
Papajik T [17]	PET/CT	Biograph 16 HIREZ, Siemens	400	60 ± 3	Visual and semiquantitative	SUV_max_
Conte MJ [18]	PET/CT	NR	NR	NR	Visual and semiquantitative	SUV_max_
Falchi L [19]	PET and PET/CT	NR	NR	NR	Visual and semiquantitative	SUV_max_
Mauro FR [20]	PET/CT	NR	NR	NR	Visual and semiquantitative	SUV_max_
Michallet AS [21]	PET/CT	NR	NR	NR	Visual and semiquantitative	SUV_max_
Mato AR [22]	PET/CT	NR	NR	NR	Visual and semiquantitative	SUV_max_
Pontoizeau C [23]	PET/CT	NR	NR	NR	Visual and semiquantitative	SUV_max_, MTV, TLG
Wang Y [24]	PET	NR	NR	NR	Visual and semiquantitative	SUV_max_
Porrazzo M [25]	PET/CT	Discovery 710, GE Healthcare	4 MBq/Kg	60 ± 10	Visual and semiquantitative	SUV_max_
Albano D [26]	PET/CT	Discovery ST or 690, GE Healthcare	3.5–4.5 MBq/Kg	60	Visual and semiquantitative	SUVbw, SUVlbm, SUVbsa, L-L SUV R, L-BP SUV R, MTV, TLG
Hod K [27]	PET/CT	GEMINI TF, Philips	185–370 MBq	60	Visual and semiquantitative	SUV_max_, SUVmean, SUV_max_/SUVliver mean ratio
Albano D [28]	PET/CT	Discovery ST or 690, GE Healthcare	3.5–4.5 MBq/Kg	60	Visual and semiquantitative	SUVbw, SUVlbm, SUVbsa, L-L SUV R, L-BP SUV R, MTV, TLG; radiomics (first- and second-order)

PET/CT: positron emission tomography/computed tomography; 2-[^18^F]FDG: fluorine-18 fluorodeoxyglucose; SUV: standardized uptake value; bw: body weight; lbm: lean body mass; bsa: body surface area; L-L SUV R: lesion-to-liver SUV ratio, L-BP SUV R: lesion-to-blood-pool SUV ratio; MTV: metabolic tumor volume; TLG: total lesion glycolysis; nr: not reported.

## Data Availability

The data presented in this study are available upon request from the corresponding author.

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
