# Peer review of "The Diagnostic Performance of 2-[18F]FDG PET/CT in Identifying Richter Transformation in Chronic Lymphocytic Leukemia: An Updated Systematic Review and Bivariate Meta-Analysis"

_cancers, 2024, doi:10.3390/cancers16091778_

Round 1

Reviewer 1 Report

Comments and Suggestions for Authors

This review is described about the correlation between SUVmax and RT in CLL patients. Even it is a strong point that two researchers searched published articles independently, I think this review might have a serious problem. 

(Major Points)

1.     It must be mentioned whether the SUVmax was only evaluated at the site of biopsy. Otherwise this report would mislead the readers completely. This is the most important point.

(Minor points)

1.     Description of the selection process of the articles is redundant. The authors should state the seven selected articles in detail.

2.     It is better that the phase “negative predictive value” in the Abstract, as well as Simple Summary.

3.     Lane 162, “Almost all” might be reworded as “Most”. 

Comments on the Quality of English Language

None. 

Author Response

(Major Points)

1. Thanks a lot for your comments and suggestions. Of course, the point of SUVmax is fundamental and it was not correctly described by us. Thus, we modified and explained better that the biopsy was performed in the site of highest uptake (highest uptake).

In fact, one of the emerging indications of 2-[18F]FDG PET/CT is detecting the best site for biopsy to recognize a potential transformation of indolent lymphomas in more aggressive variants. It's the site with higher metabolic tracer uptake that is associated with a higher risk of aggressive evolution and this is the site to biopsy.

(Minor points)

1. This is the standard study selection procedure that we follow to perform this analysis. However, we reduce the length of this part.

2. Ok correct

3. Ok correct

Reviewer 2 Report

Comments and Suggestions for Authors

In this systematic review and bivariate meta-analysis the authors analyzed the published findings about the diagnostic performance of  PET/CT in diagnosis of Richter transformation (RT). The fifteen studies that met the criteria for inclusion  with a total of 1593 CLL patients were included. The Authors confirmed  that PET/CT has a significant role  in the detection of RT in CLL patients.  The study is well written, scientifically accurate and clinically important. However, this study has several limitations. Included studies were  often retrospective and monocentric.  Moreover,  the wide heterogeneity related to several factors including their characteristics .     

Author Response

Dear Reviewer,

thank you for your comment. we are partially agree with you.

Meta-analyses offer many advantages over individual studies including:

- they have greater statistical power and the ability to extrapolate to the broader population.

- they are evidence-based articles with an original statistical analysis

- It is more likely to show an effect because smaller studies are combined into one larger study.

- they have better accuracy compared to single studies (because smaller studies are pooled and analyzed).

- they provide a view of the research that has been done in a particular field, summarize and integrate the different findings, and in particular they provide possible directions for future research.

- they also reduce the amount of work required to research a topic for other researchers and policymakers. For example, instead of having to look at the results of many smaller studies, researchers can get a more accurate view of what might be happening in a population by looking at the results of one meta-analysis.

Meta-analyses have also limitations, including heterogeneity, that should be always reported.

As currently there is not a published meta-analysis on the selected topic, we believe that this article could add relevant information compared to the available literature, for the reasons listed above.

Reviewer 3 Report

Comments and Suggestions for Authors

I have reviewed the manuscript. Although the manuscript is scientifically merit, I do not think that this meta-analysis does add anything new, to what we have already known on this topic.

Comments on the Quality of English Language

There are minor grammatical errors and typos that need to be checked and corrected.

Author Response

Dear Reviewer,

thank you for your comment. we are partially agree with you.

Meta-analyses offer many advantages over individual studies including:

- they have greater statistical power and the ability to extrapolate to the broader population.

- they are evidence-based articles with an original statistical analysis

- It is more likely to show an effect because smaller studies are combined into one larger study.

- they have better accuracy compared to single studies (because smaller studies are pooled and analyzed).

- they provide a view of the research that has been done in a particular field, summarize and integrate the different findings, and in particular they provide possible directions for future research.

- they also reduce the amount of work required to research a topic for other researchers and policymakers. For example, instead of having to look at the results of many smaller studies, researchers can get a more accurate view of what might be happening in a population by looking at the results of one meta-analysis.

Meta-analyses have also limitations, including heterogeneity, that should be always reported.

As currently there is not a published meta-analysis on the selected topic, we believe that this article could add relevant information compared to the available literature, for the reasons listed above.

Moreover, typos and spelling errors are corrected

Round 2

Reviewer 1 Report

Comments and Suggestions for Authors

Well revised. I have no more comments.